# Systematic Evaluation of Permeability of Concrete Incorporating Coconut Shell as Replacement of Fine Aggregate

**DOI:** 10.3390/ma15227944

**Published:** 2022-11-10

**Authors:** Akram M. Mhaya, Hassan Amer Algaifi, Shahiron Shahidan, Sharifah Salwa Mohd Zuki, Mohamad Azim Mohammad Azmi, Mohd Haziman Wan Ibrahim, Ghasan Fahim Huseien

**Affiliations:** 1Faculty of Civil Engineering and Built Environment, Universiti Tun Hussein Onn Malaysia, Parit Raja 86400, Batu Pahat, Johor, Malaysia; 2School of Civil Engineering, Faculty of Engineering, Universiti Teknologi Malaysia, Skudai 81310, Johor Bahru, Johor, Malaysia; 3Institute of Architecture and Construction, South Ural State University, Lenin Prospect 76, 454080 Chelyabinsk, Russia

**Keywords:** waste materials, cleaner concrete, durability, coconut shell, systematic experimental work, informational modeling

## Abstract

The concern about coconut shell disposal and natural fine aggregate depletion has prompted researchers to utilize coconut shell as aggregate in recent years. However, the majority of the present literature has focused on utilizing coconut shell as a coarse aggregate replacement in concrete via the traditional method. In this study, concrete incorporating coconut shell as a fine aggregate replacement (10–100%) was evaluated using permeability and water absorption tests in a systematic way. The response surface methodology (RSM) was first used to design the experimental works. In addition, an artificial neural network (ANN) and genetic expression programming (GEP) were also taken into account to mathematically predict the permeability and water absorption. Based on both experimental and theoretical modeling, three scenarios were observed. In the first scenario, high quality concrete was achieved when the replacement percentage of sand by coconut shell ranged from 0% to 10%. This is because both the permeability and water absorption were less than 1.5 × 10^−11^ m and 5%, respectively. In the second scenario, an acceptable and reasonable low permeability (less than 2.7 × 10^−11^ m/s) and water absorption (less than 6.7%) were also obtained when the replacement percentage increased up to 60%. In contrast, the high content coconut shell, such as 90% and 100%, developed concrete with a high permeability and water absorption and was defined in the third scenario. It was also inferred that both the experimental and mathematical models (ANN, GEP, and RSM) have consistent and accurate results. The correlation statistics indicators (*R*^2^) were greater than 0.94 and the error was less than 0.3, indicating a strong correlation and minimum error. In conclusion, coconut shell could act as a good alternative material to produce cleaner concrete with an optimum value of 50% as a fine aggregate replacement.

## 1. Introduction

Coconut shell is one of the agricultural wastes that becomes a contributor to environmental pollution [1,2,3,4,5]. It is grown in more than 90 countries in areas measuring 14.231 million hectares with a total copra production equivalent to 11.04 million tonnes. Indonesia (25.63%), the Philippines (23.91%), and India (19.20%) are the major coconut-producing countries in the world [6,7]. In 2018, it was reported that a total of 504,700 metric tonnes of coconuts in Malaysia were produced in a year with a coverage plant area of 84,600 hectares [8,9]. As a result, researchers shifted their attention to solve this problem. Similarly, many researchers were also prompted to exploit other waste materials such as tire rubber [10,11,12], coal bottom ash [13,14,15], and glass powder [16,17,18] as a replacement of concrete aggregate. However, the target waste materials must satisfy the requirement of both the mechanical and durable qualities of concrete during the lifespan of concrete structures [19,20,21]. Permeability and water absorption tests are important indicators of concrete durability that must be taken into account [22,23,24]. This is because high permeability and water absorption could accelerate the deterioration of the concrete matrix. In particular, it promotes the easy path of aggressive ions to penetrate concrete and ultimately destroy the physical and structural integrity of concrete. For example, Shaaban and Rizzuto [25] evaluated the air permeability of concrete incorporating waste tire extracts. The study outcome revealed that the inoculation of 1% steel fibers and 10% crumb rubber led to the improved air permeability of concrete exposed to oven-drying compared to that of the air-dried specimens [26]. The improvement of permeability was attributed to pores filling due to rubber expanding at high temperatures. Moreover, the permeability of self-compacting concrete minimized with the addition of natural waste perlite powders (260 kg/m^3^) in a study reported by El Mir and Nehme [27]. Similarly, exploiting polished granite waste as a partial replacement of coarse aggregate exhibited a better result for both water permeability and water absorption [28,29]. In contrast, the water permeability of concrete incorporating waste glass as a partial replacement of sand increased owing to the development of extra voids between cement paste and waste glass particles at the interface [30]. In the same context, the combination of both coconut shell and fly ash as partial replacements of coarse aggregate and cement, respectively, improved the resistance against water absorption and permeability in a study reported by Prakash and Thenmozhi [31]. Similarly, the replacement of coarse aggregate by coconut shell in self-compacting concrete exhibited positive results up to 75% in the presence of silica fume and rice husk ash [32,33,34].

From another perspective, the majority of the existing literature relies on the traditional lab work, while the mathematical and systematic experimental work of concrete containing coconut shell is still in its infancy. In this spirit, extensive theoretical research was carried out to predict and optimize the concrete durability using response surface methodology (RSM), artificial neural networks (ANN), and genetic expression programing (GEP) in the existing literature [35,36,37,38]. Indeed, mathematical modeling is regarded as one of the important methods not only to minimize the number of experiments but also to carefully assess the relationship and interaction between variables. For instance, Abbas [39] developed a non-linear model to predict and optimize the permeability of high performance concrete incorporating silica fume and natural pozzolan. Similarly, Güneyisi and Gesoğlu [40] improved the optimum content of metakaolin and fly ash that achieved the best performance of chloride permeability and water absorption of concrete using the response surface method. Moreover, the statistical algorithms method was also adopted to predict the chloride permeability of self-compacting concrete in a study reported by Kumar and Rai [41]. In addition, the evolved support vector regression method proved its ability to effectively predict the permeability coefficient of pervious concrete in a study reported by Sun and Zhang [42]. Based on the study’s findings, a high correlation coefficient (*R*) and a low root-mean-square error (RMSE) were achieved indicating that the model was reliable and efficient [43]. 

It can be inferred that very limited experimental research was conducted to examine the permeability and water absorption of concrete containing coconut shell as fine aggregates. Accounting for that in this present study, and to fill the gap of the existing literature, both experimental and mathematical modeling were considered to predict and obtain the optimum content of coconut shell as the replacement of fine aggregate using genetic expression programming (GEP), artificial neural networks (ANN), and response surface methodology (RSM). In addition, ANOVA, error and correlation statistical validation methods were considered to evaluate the performance of the proposed models.

## 2. Experimental and Informational Modeling

As shown in Figure 1, phase I dealt with determining the experimental results of the proposed concrete in terms of water absorption and permeability. Phase II dealt with the assessment of the theoretical analysis of the proposed concrete.

### 2.1. Experiment Design

Experiment design is an optimization algorithm that involves both mathematical and statistical procedures to analyze the interaction and relationship between the output or, namely, the responses (dependent variables) and input or influential factors (independent variables) [44]. In addition, the two main ideas behind the exploit of experiment design are (1) determine a minimum number of experimental tests and (2) obtain the optimum values of the influential factors that achieve the best performance of the target output. The central composite design (CCD) of response surface methodology is a type of experiment design approach that is used for optimization purpose in the present study. Indeed, CCD is composed of three types of points including center points, 2*n* axial points and 2*^n^* factorial points where *n* is the number of independent variables. Figure 2 shows the required experimental tests on the basis that two independent variables were used. It is interesting to note that the coded values +1 and −1 refer to the high and low limit of each parametrize, while α is the distance from center of the cube which equal to 1.414 in the present study. It should be also noted that five center points were adopted to examine and assess the prediction error. Similarly, the number of experimental tests (*Q*) was found to be thirteen using Equation (1) where *m* represented the number of center points [45]. Moreover, Equation (2) was used to convert the coded values to real values where *Z* and *Zc* are the real value of the independent value and real value of independent variables at the center point, respectively [46]. Furthermore, *L* denoted the coded value of the independent variable. (1)$$Q=2^{n}+2n+m$$
(2)$$L=\frac{Z-Z_{c}}{\alpha }$$

A second order polynomial equation was considered to represent the permeability of CA-based concrete. The general form of the second order polynomial equation is presented in Equation (3) [47] where *β*_ii_ is the quadratic coefficients, *β_o_* corresponds to intercept of the model, and *β*_i_ denotes to the linear coefficients. Furthermore, *X*_1_ and *X*_2_ represent the input data involving CA content and time, while *Y* is the response (permeability and water absorption). For the purpose of verification of the proposed equation, analysis of variance (ANOVA) was taken into account. In particular, *R*^2^ was calculated to evaluate the closeness between the response and real results as shown in Equation (4) where *SS_T_* represents the total sum of square error, *SS_E_* is the sum of square error based on the predicted results, and ${\bar{Y}}_{A}$ denote the mean value of actual value. In addition, *Y_P_* and *Y_A_* are the predicted and real values. In the same context, $R_{adj.}^{2}$ was also calculated to evaluate the effect of the number of independent variables on the correlation between the real and predicted results as shown in Equation (5) [48] where *DF* is the degree of freedom and *SS_R_* represents the sum square of differences error between the actual and predicted values. The predicted $R_{pred.}^{2}$ was also determined according to Equation (6) [49]. The differences between $R_{pred.}^{2}$ and $R_{adj.}^{2}$ should be less than 0.2 to ensure that the equation has the ability to predict for more data [50] where *W* refers to the estimated residual sum of square without the *i*th. Meanwhile, the signal-to-noise ratio was evaluated using the adequate precision (*SN*) as shown in Equation (7) in which its value should be greater than 4 [51] where σ^2^ denote to the residual mean square. In addition, the *p*-value and F-value were also taken into account to validate the significance of the proposed equation. The achievement of the high F-value and *p*-value less than 0.005 led to the equation being considered as significant [52].
(3)$$Y=\beta_{o}+\sum \beta_{i}X_{i}+\sum \beta_{ii}X_{i}^{2}+\sum \beta_{ij}X_{i}X_{j}$$
(4)$$\begin{array}{ll} R^{2} & =\frac{SS_{E}}{SS_{T}}\\ & =\frac{{\sum_{i=1}^{n}\left(Y_{P}-{\bar{Y}}_{A}\right)}^{2}}{{\sum_{i=1}^{n}\left({\bar{Y}}_{A}-Y_{A}\right)}^{2}} \end{array}$$
(5)$$R_{adj}^{2}=1-\frac{SS_{R}/DF_{R}}{SS_{T}/DF_{T}}$$
(6)$$R_{pred}^{2}=1-\frac{W}{SS_{T}}$$
(7)$$SN=\frac{\max(Y_{p})-\min(Y_{p})}{\sqrt{\frac{p\sigma^{2}}{n}}}$$

### 2.2. Preparation of Concrete Mix Design

Prior to obtaining the ingredients of the concrete mixture, the materials are first collected and prepared to satisfy the specification. For example, the crushed granite was exploited as coarse aggregate, while the fine aggregate was collected for the natural sand. Both the coarse and fine aggregate met the requirement of international standard EN 933-1:2012. Meanwhile, the ordinary Portland cement is produced (OPC). Another crucial element in the mixing of concrete is water. Water helps to bind the cement and aggregates to produce concrete. BS EN 1008: 2002 defines sources of water and provides requirements as well as testing frequencies for qualifying individual or combined water sources. Apart from that, the pH value of water is also taken into consideration. In this research, the source of water was tap water used for water absorption and permeability tests. Regarding coconut shell, it was collected from a local supplier in Malaysia. After that, the CA is exposed to grinding via an impact pulverizer to achieve the suitable target size (Figure 3a). The sieve analysis was also considered to ensure that all CA particles passed a sieve size of 4.75 mm. In addition, a sufficient amount of CA particles passing the sieve size of 600 µm was also checked (Figure 3b). Consequently, the Department of Environment (DoE) method (British Standard) was adopted to calculate the proportions of the control concrete mixture that has strength of 30 MPa at 28 days. The obtained results from DOE revealed that the amount of cement, water, coarse aggregate, and fine aggregate were 435 kg, 195 L, 1251 kg/m^3^ and 536 kg/m^3^, respectively. It is interesting to note that the CA was added as a partial replacement of fine aggregate according to the suggested array experiment design of CCD as shown in Table 1. Rheobuild 1100 superplasticising admixture (up to 1.2% by cement weight) was also used to acquire a slump in the range of 100–140 mm. Table 2 exhibits the mix proportion of the proposed concrete.

### 2.3. Water Absorption Test

Aiming to explore the impact and extent to which coconut shell could affect the capillary and concrete voids, a water absorption (*WA*) test was carried out in compliance with BS 1881-122:2011. Three concrete cubes with the dimension of (100 × 100 × 100) mm were considered for each experiment run as shown in Figure 4. All cubes were tested after 28 days of curing age. Moreover, the value of water absorption was calculated using a traditional mathematical formula as described in Equation (8) where *W*_2_ and *W*_1_ represent the wet weight of the concrete cube and the dry weight of the concrete cube, respectively. It is also interesting to note that the correction factor (*CF*) was taken into account to tackle the sample length variation, which was in line with the study reported by Kwan and Ramli [53]. (8)$$WA=\frac{W_{2}-W_{1}}{W_{1}}\times 100\times CF$$

### 2.4. Permeability Test

To further analyze the evolution of water transport in concrete with and without coconut shell, a water permeability test was conducted in accordance with BS EN-12390-8:2009. In particular, water permeability coefficients (*K*_w_) were obtained to evaluate the performance of concrete incorporating CA as shown in Equation (9) where *d* is the depth of water penetration, *T* refers to time under pressure, and *h* represent the hydraulic head. In addition, the porosity (*v*) is the function of the area of the cubes (*A*), water density (ρ), depth of penetration (*d)*, and the differences of mass sample (*m*) as shown in Equation (10). To implement the test, the concrete cubes in the dimension of (100 × 100 × 100) mm were tested after 28 days for each experiment run. After the test arrangement was pressurized with 5 bars for 3 days as shown in Figure 4b, the concrete cubes were split into half and the water penetration depth measured. (9)$$K_{w}=\frac{d^{2}v}{2Th}$$
(10)$$v=\frac{m}{Ad\rho }$$

### 2.5. Prediction Model Using ANN

In recent years, extensive theoretical research was adopted to quickly analyze the behavior of concrete properties under different influential parameters. Herein, a data-driven model using an artificial neural network was utilized to provide a predictive mathematical equation that is able to examine and evaluate the effect of coconut shell on permeability as well as the absorption of concrete. Table 2 shows the collected real results that were inserted as input data in MATLAB to develop the ANN model. In particular, 25% of the input data were used for validation purposes, while 75% were used for training. Meanwhile, the TanH sigmoid function is taken into account as an activation function shown in Equation (11). Moreover, three layers were used to develop the data-driven model. The input data were inserted, weighted, and summed in the first layer, which simulates the collecting and receiving of the information by a human brain. After that, the input data were exposed to mathematical processes through the TanH sigmoid activated function in the second layer or namely the hidden layer. This step is line with the human process in which the received information is processed and converted to electrical signals. In the third layer, namely output layer, the action is taken either to accept or reject the output results (signal electric). In the human brain, the acceptance or rejection is dependent on the strength of the electrical signal, while it depends on the satisfaction of statistical indicators of the ANN model. In this study, the root average squared error (RASE) and *R*^2^ were adopted as statistic validation indictors to verify the accuracy and strength of the proposed equation as shown in Equations (12) and (13), respectively.
(11)$$f_{\left(x\right)}=\frac{e^{2x}-1}{e^{2x}+1}$$
(12)$$\begin{array}{ll} RASE & =\sqrt{\frac{SSR}{n}}\\ & =\sqrt{\frac{\sum_{i=1}^{n}{\left(Y_{A}-Y_{P}\right)}^{2}}{n}} \end{array}$$
(13)$$R^{2}=1-\frac{{\sum_{i=1}^{n}\left(Y_{A}-Y_{P}\right)}^{2}}{\left({\sum_{i=1}^{n}\left(Y_{A}\right)}^{2}-\frac{{\left(\sum_{i=1}^{n}Y_{A}\right)}^{2}}{n}\right)}$$

### 2.6. Prediction Model Using GEP

The theoretical framework of concrete properties has become a hot trend in the scientific community, specifically for civil structural engineers. Moreover, many researchers have recognized the GEP model as a reliable and robust model. In this study, two mathematical equations were developed to simulate the permeability and water absorption of CA-based concrete. As shown in Table 2, the collected input data were used to develop the GEM model. Two independent variables were considered involving coconut shell content and time, while two dependent variables represent the permeability and water absorption of concrete in the proposed GEP models separately. In addition, the GeneXpro Tools 5.0 software was used to achieve this goal. In a similar manner to the ANN model, the collected data were divided into groups. The first group was defined as a training step that use 75% of the data, while 25% was related to the validation step. For validation, the statistic indicators are divided into correlation and error statistics indicators. Regarding the correlation statistics indicators, *R* and *R*^2^ were calculated to assess the relationship between the real and predicted results as descried in Equations (14) and (15). The closer the *R* to one, the strength and closeness results could be achieved. For the error statistic indicators, four methods were taken into account involving the mean absolute error (*MAE*), mean root relative squared error (*RRSE*), relative absolute error (*RAE*), and root mean square error (*RMSE*) as show in Equations (14)–(19).
(14)$$R^{2}=1-\frac{\sum_{i=1}^{n}{\left(Y_{A}-Y_{P}\right)}^{2}}{\sum_{i=1}^{n}(Y_{A}-\bar{Y_{A}}{)}^{2}}$$
(15)$$R=\sqrt{1-\frac{\sum_{i=1}^{n}{\left(Y_{A}-Y_{P}\right)}^{2}}{\sum_{i=1}^{n}(Y_{A}-\bar{Y_{A}}{)}^{2}}}$$
(16)$$MAE=\frac{1}{n}\sum_{i=1}^{n}\left|Y_{A}-Y_{P}\right|$$
(17)$$RRSE=\sqrt{\frac{\sum_{i=1}^{n}{\left(Y_{P}-Y_{A}\right)}^{2}}{\sum_{i=1}^{n}{\left(Y_{A}-\bar{Y_{A}}\right)}^{2}}}$$
(18)$$RAE=\frac{\sum_{i=1}^{n}\left|Y_{A}-Y_{P}\right|}{\sum_{i=1}^{n}\left|Y_{A}-\frac{1}{n}\sum_{i=1}^{n}Y_{A}\right|}$$
(19)$$RMSE=\sqrt{\frac{1}{n}\sum_{i=1}^{n}{\left(Y_{A}-Y_{P}\right)}^{2}}$$

## 3. Result and Discussion

### 3.1. Parametric Analysis

The evolution of water absorption and permeability of concrete under different replacements of fine aggregate by coconut shell were also investigated and evaluated using both experiments and modeling involving RSM, ANN, and GEP models.

#### 3.1.1. Water Absorption

As is well known, the water absorption (*WA*) of concrete is one way to assess its durability and it is related to the movement of liquid such as water into the concrete matrix owing to surface tension acting in the capillaries. BS 1881-122:1983 defined the *WA* as the weight’s increment by absorbed water that might be occurred compared to that of the dry concrete samples. This increment is attributed to the high pressure inside the pores when the dry concrete samples are exposed to and submerged in water. Therefore, a high percentage of pore volume and interconnectivity pores inside the concrete could be indicated by the value of the water absorption of concrete samples. Less water absorption refers to dense and high-quality concrete, while high absorption could be considered as a concern of concrete deterioration. An excellent water absorption of concrete could be achieved when its value is lower than 5% according to ASTM C 642-06 [54]. 

In the present study, an excellent water absorption (less than 5%) was obtained when the replacement of fine aggregate by coconut shell was 10% compared to the normal concrete as shown in Figure 5. In the same context, it can be seen that the *WA* increased with the increase in the sand replacement level through two scenarios. In the first scenario, the increment of water absorption continued and slightly increased up to 55% of the sand replacement percentage by coconut shell. After that, the *WA* has a sharp slop and significantly increased in comparison with the normal concrete, which is defined as the second scenario. For example, the increment of water absorption was in the range of 5.48–6.63% when the sand was replaced by the coconut shell from 20% to 50%. Such results could indicate a good water absorption. This is in line with Mo and Thomas [55] who highlighted that the concrete could be classified as good quality concrete if the *WA* is less than 10%. Similarly, Shafigh and Nomeli [56] defined the concrete incorporating the partial replacement of fine aggregate (37.5%) by oil palm shell as a good quality concrete due to the reasonable value of water absorption obtained (less than 10%). Back to the second scenario, the 100% replacement of fine aggregate by coconut shell lead to an increase in the *WA* up to 13.85%, which is considered as high and not desirable water absorption. Such an increment might be attributed to the capacity of coconut shell to absorb water itself and create a high osmosis pressure inside the concrete matrix. Moreover, the coconut shell is not dense compared to that of the natural aggregate, such as granite, in which it somewhat has pores that correspond to high concrete absorption. In particular, it could increase the chance of interconnectivity pores inside the concrete microstructure when the higher replacement of sand by coconut shell was used.

#### 3.1.2. Water Permeability

Despite both the permeability and absorption of concrete being experimental indicators to assess the quality of concrete in terms of water penetration, absorption tests differ from concrete permeability tests and do not necessarily fully relate the absorption to the permeability [57]. As is well known, the microstructure of concrete is composed of a complex pore system, such as pores and interconnectivity pores, as well as invertible microcracks. Both the pores and interconnectivity are attributed to the water cement ratio, degree of compaction, and degree of hydration, while the occurrence of microcracks is invertible due to external and internal stress such as shrinkage, bleeding, and other factors that cause volumetric changes [58]. The pores are classified into three types according to its size (i.e., gel pores, capillary pore, and air voids). The size of the gel pores and capillary pores are 0.5–10 nm [59] and 50 nm–10 µm, respectively [60], whereas the size of air voids ranges from tens of micrometers to a millimeter [61]. These pores will later connect with each other to form an interconnectivity pore system that is considered as the main idea behind the permeability concept. In other words, the continuity of pores, their size, and distribution are related to the permeability. As such, it can be inferred that the permeability of concrete is important as it negatively causes the lifespan of cement-based structures to be exposed to water or aggressive chemical ions. 

In the present study, the water permeability of control concrete without the coconut shell was 1.1 × 10^−11^ m/s at 28 days, which is in a good agreement with the present literature. For example, Cuadrado-Rica and Sebaibi [62] concluded that the range of normal concrete permeability was 1.0 × 10^−11^ m/s to 5.0 × 10^−11^ m/s. In addition, according to ACI standard 301-89, a high-quality concrete is obtained when the water permeability is lower than 1.5 × 10^−11^ m/s. In our study, the water permeability of concrete containing 10% of the coconut shell as a replacement of the fine aggregate was found to be 1.2 × 10^−11^ m/s confirming that the concrete quality is still high when it met the specifications. In the same context, with the increase in the coconut shell up to 60%, the water permeability of the concrete slightly increased up to 2.7 × 10^−11^ m/s, which also can be considered a low and reasonable permeability. This fact is in line with Amriou and Bencheikh [63], who defined a low water permeability of concrete as when its value was located in the range between 8 × 10^−12^ m/s and 3.2 × 10^−11^ m/s. This result was consistent with all the data sets obtained from the RSM, ANN, and GEP models as shown in Figure 6a. It can be seen that the permeability of concrete containing coconut shell up to 55% was lesser than 2 × 10^−11^ m/s in indicating that the concrete quality is good. After that, the slop of water permeability increased with the increases in coconut shell content, specifically, 90% and 100%. This result is also presented using a counter plot as shown in Figure 6b–d in which the blue color reflects low permeability, whereas the yellow and red colors relate to high permeability. This mean that the zone of low permeability was located between 0 and 60% of coconut shell replacement, while the incorporation of high content coconut shell (more than 60%) would weaken the resistance of the concrete permeability. In addition, it might be attributed to the increment of pores and interconnectivity pores of concrete incorporating the high content of coconut shell. 

### 3.2. Informational Modeling Using RSM

The evolution of the water permeability and absorption of CA-based concrete were estimated and evaluated using quadratic equations as shown in Table 3. It can be seen that both equations are functions of time and coconut shell content. Based on the ANOVA results, both the permeability and water absorption equations were found to be significant in which the F-values were 384.41 and 300.17, while the *p*-values were 0.0026 and 0.0033, respectively. This is consistent with the previous studies that determined that the model could be considered as significant if the *p*-value is lesser than 0.005 and F-value is high [64]. The accuracies of both equations were also tested using *RMSE*. It was found that the value of the RMSE is minimum, specifically the RMSE value of *WA* and *WP* were 0.1492 and 8.2 × 10^−13^, indicating that the model was able to estimate accurate results. This fact is in line with Algaifi and Alqarni [65] who used RMSE to prove the adequacy of the predicted equation of bacterial concrete strength. Based on their outcomes, the RSME was 2.04, confirming that the predicted and actual results were close and accurate. In the same contest, *R*^2^ proved the closeness and correlation between the predicted and actual results in which the value of *R*^2^ was high. According to Huseien and Sam [66], a good correlation could be obtained when *R*^2^ is greater than 0.7. Herein, the *R*^2^ value of *WA* and *WP* were 0.9993 and 0.9995, thus highlighting that the predicted results were acceptable. In addition, the capability of these quadratic equations to accurately predict further data was also proved using $R_{predicted}^{2}$ and $R_{adj}^{2}$. In particular, it was found that the differences between the $R_{predicted}^{2}$ and $R_{adj}^{2}$ were less than 0.2. This fact is in good agreement with the existing literature. For example, Jitendra and Khed [67] developed an RSM model to predict and optimize the water absorption, chloride ions penetration, and compressive strength of concrete blocks containing foundry sand and fly ash as partial replacements of natural sand and cement, respectively. The outcome of their study revealed that the model was reliable and could be used for further prediction. This is because a reasonable difference (less than 0.2) between $R_{predicted}^{2}$ and $R_{adj}^{2}$ was achieved for all data sets. 

For the same regards, the significance of each parameter involving time and coconut shell on the permeability and water absorption of concrete was investigated and evaluated. It can be inferred that the coconut shell content has three scenarios in terms of slop gradient as shown in Figure 7a,b. In the first scenario, the line is almost horizontal indicating that the there is no effect. This is because the replacement percentage of CA is still low (less than 25%). After that, the line slop started to increase up with the increase in the CA content up to 50% which was classified and defined in the second scenario. In the third scenario, a sharp slop was observed indicating that the CA content (higher than 50) has a great significance on both the water absorption and permeability. In particular, with the increase in coconut shell content, both the permeability and water absorption were also increased. This fact is also confirmed by ANOVA in which the *p*-value of the coconut shell content was 0.0007 and 0.0006 based on the water absorption and permeability, respectively, while the F-value of *CA* was 1352.83 and 1787.27. 

Moreover, the desirability functions were also used to optimize the optimum value of coconut shell content in CA-based concrete. Equation (19) represents the general mathematical form for the optimization purpose using RSM. Herein, the optimization equation is a function of two independent variables involving time and CA content. It was found that 78 solutions were obtained using the optimization equation. In addition, the optimal content of the CA content was considered as 53% in the present study as shown in Figure 8a. This is because, beyond this value, the increment of water absorption and permeability greatly increased. This fact is also in line with the illustrated results in Figure 8b,c. For example, two different slops were recorded. In the first slop, the increment of WA and WP of concrete incorporating CA up to 55% was reasonable and almost acceptable, while the high replacement percentage of CA (greater than 55%) significantly increased the WA and WP. This is also consistent with BS 1881-122 (2011) in which a good quality concrete could be considered when the concrete absorption is lower than 10%. The second reason to consider 53% as an optimum value is that a significant reduction on the mechanical properties of concrete was recorded when the CA content is greater than 53%, which was explained and discussed in our previous published paper. From another point of view, our attention focused to produce normal concrete incorporating CA. Beyond 53% of the CA, the concrete could be classified as lightweight concrete in which its density was lower than 2240 kg/m^3^. This fact is line with Khoshkenari and Shafigh [68] and Nowak and Rakoczy [69], who demonstrated that the density of lightweight concrete has a density in the range of 1440–1840 kg/m^3^, while the density of ordinary or normal concrete is in the range between 2240 and 2400 kg/m^3^. Herein, the concrete density is divided into two zones as shown in Figure 8d. The first zone represents the normal concrete that has a density greater than 2240 kg/m, while the second zone represent the lightweight concrete that has a lower a density.

### 3.3. Informational Modeling Using GEP and AMM

The evolution of water absorption and permeability were also theoretically predicted using GEP. As shown in Figure 9a,b, the WA and WP equations were first developed and expressed using structural trees, respectively. It can be seen that each equation has one chromosome that, in turn, composed of one gene involving (/, +, Sqrt, Avg. and constants (c_o_)). The structural tree was later converted into a mathematical expression using the Karva language.

The accuracy and reliability of the predicted GEP equations were also examined and assessed using both correlation and error statistics validations methods. It was found that the differences between actual and estimated results were found to be minimum and acceptable. According to Ali Khan and Zafar [70], the accuracy of the predicted equation of geopolymer concrete was proved using RMSE and MAE values lower than 5.971 and 5.832 for both training and validation. Herein, the values of RMSE and MAE were less than 1.037 for both WA and WP equations, indicating that the predicted GEP results could be considered as accurate and reliable. Similar trends were also observed using an ANN model. It was found that the RSME value of WA and WP of CA-based concrete was lower than 0.4447 for both training and validation data as shown in Table 4. Such results confirmed that both ANN and GEP equations could be used for further prediction with less errors. In the same context, high correlation and closeness between the actual and predicted results were obtained for both ANN and GE models. Based on Alabduljabbar and Huseien [71], the good correlation between the predicted and actual results of the alkali activated concrete’s compressive strength were obtained using *R*^2^ values of 0.991 and 0 and 9878 for training and validation. This outcome was similar to the present findings. In particular, the value of *R*^2^ of the proposed water permeability equation using GEP and ANN were 0.9519 and 0.9719 for training, while its value was higher than 0.967 for validation. These positive results are almost in line with the findings from the predicted equation of water absorption using ANN and GEP. In particular, the *R*^2^ were higher than 0.856 for both training and validations. As such, it can be inferred that both GEP and ANN models showed its ability to predict with high correlation and minimum error. 

## 4. Conclusions

In recent years, the utilization of waste agricultural material as an alternative of aggregate in concrete was rapidly increased in order to address the environmental problems. In the present study, coconut shell was taken into account as a replacement of fine aggregate. Both the permeability and water absorption of CA-based concrete tests were assessed in a systematic way using RSM, ANN and GEP models. It can be concluded that utilizing coconut shell as a fine aggregate is helpful to address environmental problems, however, our work was focused on conventional concrete. Pozzolanic-based material such as fly ash could be very useful to enhance and produce CA-concrete. In addition, further study is encouraged to assess the thermal conductivity of concrete using coconut shell. 

In the same context, based on the experimental and predicted results, it can be concluded that:All mathematical models of RSM, ANN, and GEP proved their ability to evaluate the behavior of CA-based concrete, in which the predicted data and the actual data were consistent.Based on ANN, GEP, and RSM models, the replacement percentage of fine aggregate by coconut shell up to 50% produce a good quality concrete in which the permeability and water absorption were less than 2.7 × 10^−11^ m/s and 5%, receptively.The ANN, RSM, and GEP also revealed that the high replacement of fine aggregate by coconut shell produced a concrete with high permeability (greater than 4.5 × 10^−11^ m/s) and high water absorption (greater than 10%).

## Figures and Tables

**Figure 1 materials-15-07944-f001:**
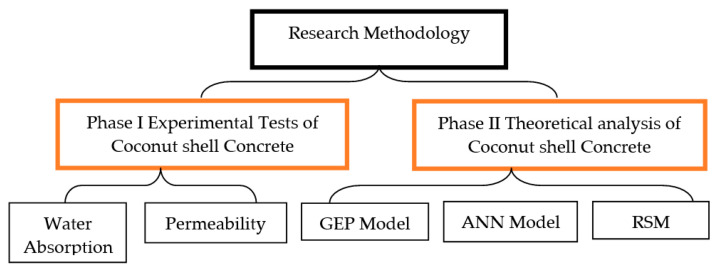
Diagram of representation of empirical program.

**Figure 2 materials-15-07944-f002:**
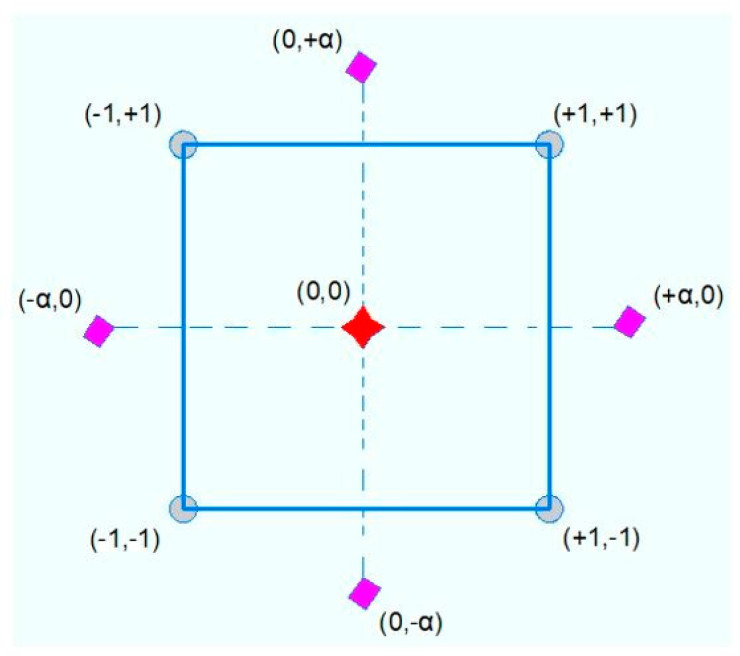
Basic concept of experimental design use.

**Figure 3 materials-15-07944-f003:**
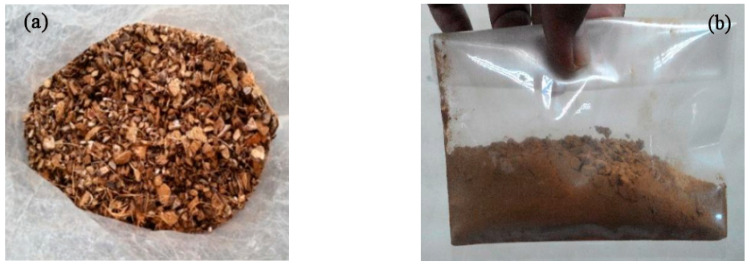
Preparation of coconut shell: (**a**) chips (20 mm) and (**b**) fine powder.

**Figure 4 materials-15-07944-f004:**
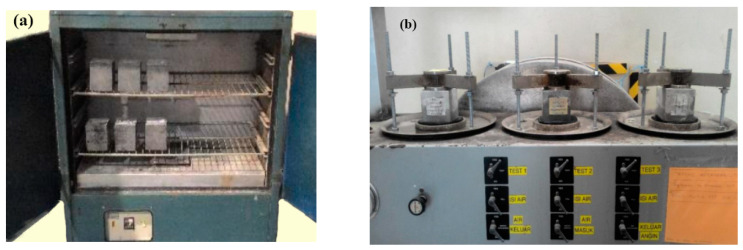
Water penetration tests set up: (**a**) water absorption and (**b**) water permeability.

**Figure 5 materials-15-07944-f005:**
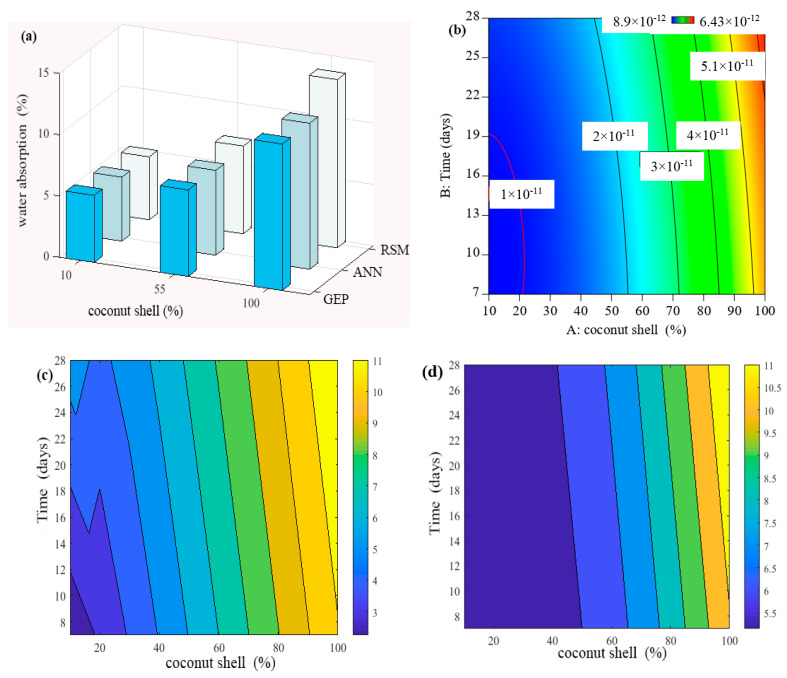
Evolution of water absorption of CA-based concrete: (**a**) all models (**b**), RSM (**c**) GEP, and (**d**) ANN model.

**Figure 6 materials-15-07944-f006:**
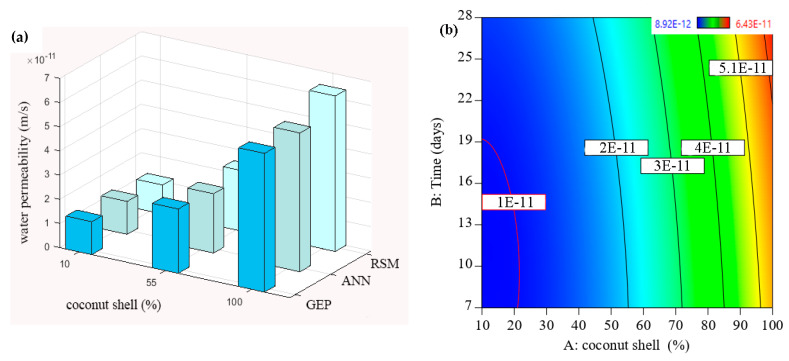
Evolution of water permeability of CA-based concrete: (**a**) all models, (**b**) RSM, (**c**) GEP, and (**d**) ANN model.

**Figure 7 materials-15-07944-f007:**
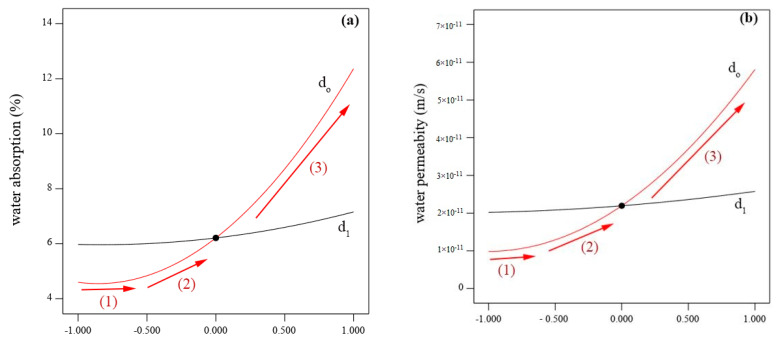
Significance of the involved influential parameters on: (**a**) water absorption and (**b**) permeability.

**Figure 8 materials-15-07944-f008:**
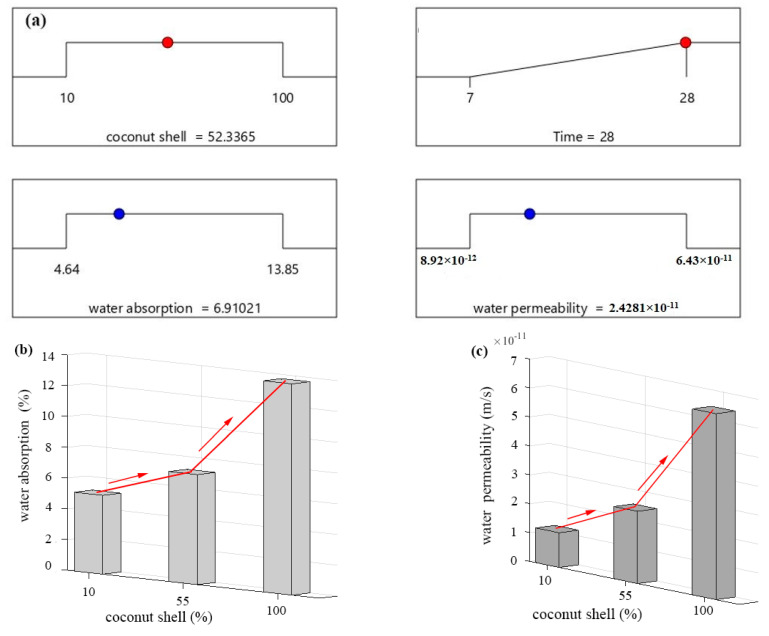
Optimization of coconut shell: (**a**) desirability functions, (**b**) predicted WA evolution, (**c**) predicted WP evolution, and (**d**) CA-based density evolution.

**Figure 9 materials-15-07944-f009:**
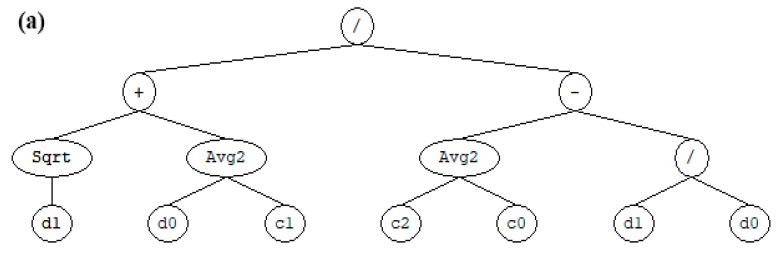
Structural tree of the predicated behavior of CA-based concrete: (**a**) water absorption and (**b**) water permeability.

**Table 1 materials-15-07944-t001:** The required experimental run according to CCD.

Run NO.	Coded Value		Real Value		CCD Division
			Replaced Coconut Shell (%)	Time (Days)	
1	−1	−1	10	7	Factorial points (2*^n^*)
2	1	−1	100	7	
3	−1	1	10	28	
4	1	1	100	28	
5	1	0	100	17	Axial points (2*n*)
6	−1	0	10	17	
7	0	−1	55	7	
8	0	1	55	28	
9	0	0	55	17	Centre points
10	0	0	55	17	
11	0	0	55	17	
12	0	0	55	17	
13	0	0	55	17	

**Table 2 materials-15-07944-t002:** The actual experimental data used for ANN and GEP models.

Name of Specimens	Percentage of Fine Coconut Shell	Permeability m/s		Water Absorption %	
		7 Days	28 Days	7 Days	28 Days
FCSC10	10%	8.92 × 10^−12^	1.2 × 10^−11^	4.65	5.21
FCSC20	20%	1.37 × 10^−11^	1.53 × 10^−11^	5.12	5.48
FCSC30	30%	1.48 × 10^−11^	1.73 × 10^−11^	5.34	5.53
FCSC40	40%	1.81 × 10^−11^	1.96 × 10^−11^	5.96	6.28
FCSC50	50%	2.02 × 10^−11^	2.3 × 10^−11^	5.99	6.63
FCSC60	60%	2.17 × 10^−11^	2.7 × 10^−11^	6.37	7.28
FCSC70	70%	2.58 × 10^−11^	3.38 × 10^−11^	7.78	8.17
FCSC80	80%	3.59 × 10^−11^	3.56 × 10^−11^	8.45	8.77
FCSC90	90%	4.52 × 10^−11^	4.69 × 10^−11^	9.54	10.23
FCSC100	100%	5.47 × 10^−11^	6.43 × 10^−11^	11.63	13.85

FCS a coconut shell as partial replacement of sand, FCSC10 (10% fine aggregates replacement), FCSC100 (100% fine aggregates replacement).

**Table 3 materials-15-07944-t003:** Verification of the permeability and water absorption equations (RSM model).

Item	Second Polynomial Equations and the Involved Statistics Parameters				
Water Permeability (*WP*)	*R*^2^ = 0.999	$R_{adj}^{2}=$0.9964	$R_{predicted}^{2}$ 0.9813	Adeq. Precision 44.47	RMSE 8.2 × 10^−13^
	$WP=10E-12\left(0.219+0.24d_{0}+2.78d_{1}+1.63d_{0}d_{1}+0.12d_{0}^{2}+0.1d_{1}^{2}\right)$				
Water Absorption (*WA*)	*R*^2^ = 0.9987	$R_{adj}^{2}=$0.9953	$R_{predicted}^{2}$ 0.976	Adeq. Precision 40.757	RMSE 0.1492
	$WA=6.21+3.88d_{0}+0.59d_{1}+0.42d_{0}d_{1}+2.27d_{0}^{2}+0.355d_{1}^{2}$				

**Table 4 materials-15-07944-t004:** The developed ANN and GEP equations and their performance.

Model	Item	Mathematical Equation and Related Statistics Validation Parameters				
GEP	*WA*	Training	MAE = 0.817	RMSE = 1.037	*R* = 0.925	*R*^2^ = 0.856
		Validation	MAE = 0.848	RMSE = 0.959	*R* = 0.982	*R*^2^ = 0.964
		$WA=\frac{2\sqrt{d_{1}}+d_{0}+c_{1}}{-2d_{1}d_{0}^{-1}+c_{2}+c_{0}}$				
	*WP*	Training	MAE = 2.73 × 10^−12^	RMSE = 3.2 × 10^−12^	*R* = 0.9756	*R*^2^ = 0.9519
		Validation	MAE = 2.6 × 10^−12^	RMSE = 2.8 × 10^−12^	*R* = 0.995	*R*^2^ = 0.991
		$WP=10^{-12}\times \left(0.5\sqrt[{3}]{d_{1}^{2}+d_{1}-d_{0}}+0.5EXP\left(\sqrt[{3}]{d_{0}}\right)\right)$				
ANN	*WA*	Training	MAE = 0.32	RMSE = 0.4447	*R* = 0.979	*R*^2^ = 0.9598
		Validation	MAE = 0.123	RMSE = 0.221	*R* = 0.983	*R*^2^ = 0.967
		$WA=10.16-5.15\mathrm{Tan}H\left(0.5\left(4.91-0.05d_{0}-0.019d_{1}\right)\right)$				
	*WP*	Training	MAE = 2.0 × 10^−12^	RMSE = 2.4 × 10^−12^	*R* = 0.985	*R*^2^ = 0.9719
		Validation	MAE = 6 × 10^−13^	RMSE = 7 × 10^−13^	*R* = 0.99	*R*^2^ = 0.9981
		$WP=10^{-12}\times \left(45.9-33.7\mathrm{Tan}H\left(0.5\left(4.91-0.05d_{0}-0.019d_{1}\right)\right)\right)$				

## Data Availability

Data sharing not applicable.

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
