# Peer review of "Systematic Evaluation of Permeability of Concrete Incorporating Coconut Shell as Replacement of Fine Aggregate"

_materials, 2022, doi:10.3390/ma15227944_

Round 1

Reviewer 1 Report

1) Authors Can highlight the research gap and novelty of the study in a paragraph.

2)  If they highlight the potential countries where coconut shell production is high and how they will processed and procured it will add more value to the study

3) Why only few properties are concentrated  in the study?

4) Prepare one flow chart to highlight the complete process of the study

5) Reaction of coconut shell with temperature and acid impact needs to be highlighted

6) Any standards to compare the results was available ?

7) Many places the statements  are too lengthy it can be rephrased.

8) Highlight how this study is importance in Asian countries

9) Authors didn't talk about quality of the water what they used for testing

10 ) Authors needs to explain the limitations and how this study helpful to the society.

Author Response

Reviewer' comments is greatly appreciated. Attached please find the authors response to reviewer' comments. 

Reviewer 2 Report

In order to solve the problems of accumulation of agricultural waste and consumption of natural fine aggregate, the author studies cleaner concrete with coconut shell as the substitute of fine aggregate for concrete. In theory, response surface method (RSM) is used to conduct permeability and water absorption tests, evaluate the concrete with coconut shell as the substitute of fine aggregate, and predict its permeability and water absorption mathematically through artificial neural network (ANN) and genetic expression programming (GEP). Finally, it is concluded that the best substitution rate of coconut shell as a fine aggregate substitute for concrete preparation is 50%.

After reading the whole article, put forward suggestions for minor revision and acceptance. Among them, the following questions in the manuscript need to be answered and handled by the author:

1. The results of the manuscript summary show that "an optimum value of 50% as a fine aggregate replacement." in the conclusion. The conclusion is directly stated, which is somewhat abrupt. The data "an optimum value of 50% as a fine aggregate replacement" should be reflected in the results.

2. Punctuation errors occur in keywords.

3. The introduction does not reflect the necessity of using coconut shell as a substitute for fine aggregate, and the introduction and explanation of water absorption of one of the two tests in the manuscript are less. In the second paragraph of the introduction, only the GEP method is mentioned, not why the other two methods, ANN and RSM, are selected.

4. The picture is not clear enough, such as the numbers and letters in Figures 3, 5, 6 and 8. Please unify the corner mark position of the full text image. In Figure 5, 6 and 8, the number in the figure is broken or repeated. What is the corresponding text analysis? Is there any reference significance? The corresponding pictures of "2.1 Test Design" are not numbered and named.

5. The color of "14" in formula (14) number "(14)" is red, which is inconsistent with other numbers.

6. Why is the "Real value" column in Table 1 not filled in? If there is no need to fill in data, what is the significance of this column?

7. The mix proportion design of concrete is described in 2.2, but the substitution rate of coconut shell as fine aggregate is not clear. Tables should be listed to reflect different mix proportion designs, so that the expression is clearer.

8. 2.3, 2.4 After the experiment, the real data of the experiment was not reflected. How was the theoretical model derived in the later stage? Based on what data?

9. The data in Table 3 in 3.3 of the manuscript is incomplete.

10. The theoretical explanation of concrete permeability test in 3.1.2 should be simplified.

11. What are the advantages of coconut shell as fine aggregate compared with coarse aggregate? Not reflected in the conclusion. The conclusion only describes the advantages of RSM, and the advantages of the other two mathematical modeling methods are not reflected. The conclusion 3 and 4 need to be considered again.

12. The citation of references is not timely, most of which are papers before 2018, and few papers have been cited in recent two years. For example, the following documents can be referred to:

[1] Jiao Huazhe, Chen Weilin, Yu Yang*, et al. Flocculated unclassified tailings settling efficiency improvement by particle collision optimization in the feedwell, International Journal of Minerals, Metallurgy and Materials, 2022.

[2] Chen Fengbin, Xu Bin, Jiao Huazhe*, et al. Triaxial mechanical properties and microstructure visualization of BFRC[J]. Construction and Building Materials, 2021, 278(4), 122275.

Author Response

Reviewer' comments and suggestions are highly appreciated. Attached please find the authors response to reviewer' comments. 

Reviewer 3 Report

Review opinions

1. Where is Figure 1a shown in line 99?

2. The content in Table 1 is not complete. If it is not needed, do you consider deleting blank columns?

3. The picture in the article is very fuzzy. Please use the original picture in the article.

4. The experimental results of permeability and water absorption tests are similar to those of mathematical models (ANN, GEP and RSM), and good conclusions are obtained, which lays a foundation for later research.

5. Equations (9) and (10) should give the specific meaning of each physical quantity so that people can understand it better.

6. Why 50% coconut shell instead of sand can achieve more significant permeability and water absorption, the author should carry out appropriate and micro view and mechanism explanation, which will make people more convincing.

7. Conclusion (1) and (2) can not reach the conclusion effect without, whether to consider (1) and (2) in the last or combined.

Author Response

Reviewer' comments is highly appreciated, attached please find the authors response to reviewer' comments. 
